# Consumers’ Intention to Bring a Reusable Bag for Shopping in China: Extending the Theory of Planned Behavior

**DOI:** 10.3390/ijerph19063638

**Published:** 2022-03-18

**Authors:** Bairong Wang, Yong Li

**Affiliations:** 1School of Economics and Management, Shanghai Maritime University, Shanghai 201306, China; bairongw@buffalo.edu; 2School of Marxism, Shanghai Maritime University, Shanghai 201306, China

**Keywords:** reusable bags, theory of planned behavior, locus of control, environmental concern, China

## Abstract

Bringing a reusable bag for shopping is a typical pro-environmental behavior and has been shown to be effective in reducing plastics, but research regarding this green behavior is limited. In this regard, using the snowball sampling technique, this study conducts a survey on a sample of 361 Chinese consumers to investigate their intention to bring a reusable bag for shopping based on the theory of planned behavior. To increase the explaining power for behavioral intention, this study extended the TPB by adding two additional variables: locus of control and environmental concern. Data was analyzed using the structural equation modeling technique. Results show that attitude, perceived behavioral control, and subjective norm exert significant and positive influence on consumers’ intention to bring reusable bags for shopping, and the perceived behavioral control exerts the greatest influence, followed by attitude and subjective norm. Both locus of control and environmental concern fail to directly impact consumers’ bringing intention, but they could impact consumers’ intention indirectly. Specifically, the external locus of control exerts a negative influence on attitude and perceived behavioral control. Environmental concern positively impacts consumers’ attitudes towards bringing reusable bags for shopping. Results of this study could provide valuable insights into plastics management and policy design to promote consumers’ green shopping behaviors. For instance, the finding that perceived behavioral control is the greatest contributor to consumers’ intention to bring reusable bags highlights the importance of clearing reusable bag preparation and use barriers.

## 1. Introduction

Plastic pollution has caused great environmental threats all over the world [1]. Extensive efforts have been made by international governments to combat this crisis by introducing different plastic ban policies and encouraging green and sustainable shopping behaviors of consumers. Among the green shopping behaviors, bringing a reusable bag for shopping is a favorite option to reduce consumers’ usage of single-use plastic carrier bags. However, the existing studies paid great attention to consumers’ usage of plastic carrier bags, but studies regarding consumers’ use of reusable bags are few [2]. China, one of the world’s largest consumers of plastic bags, has also joined the battle against the plastic crisis and has introduced two national plastic ban policies to reduce consumers’ usage of plastic bags in 2008 and 2020, respectively [3,4]. Before the introduction of plastic bags, Chinese consumers used nylon or cloth reusable bags or other carrier bags for shopping. After their introduction, plastic bags started to prevail and even became a distinguished feature of modern shopping [5]. Given the alarming environmental damage caused by plastic bags, the Chinese government has increased the prices for single-use plastic carrier bags to move consumers’ shopping bags from plastic ones to the original reusable ones. Existing studies in Shanghai, China show that consumers’ use of reusable bags increases when plastic carrier bags become expensive or are forbidden for use [6,7]. However, what underlies people’s use of reusable bags has still not been covered in China. The theory of planned behavior (TPB), proposed by Fishbein and Ajzen [8] in the 1970s, evaluates psychological factors of investigated behaviors and performs well in interpreting and predicting various pro-environmental behaviors [9,10,11]. For instance, the model has also been used widely to explain and predict tourists’ environmental protection behaviors [12] and choice of eco-friendly destinations [9], recycling [13], purchase of green cars [11], energy conservation at workplaces [14], etc. The theory proposes that before actual behavior, intention has to be formed first, which could be predicted by: attitudes towards the related behavior; subjective norms, i.e., the expectation or influence from important other people, e.g., families and friends; and perceived behavioral control, i.e., the perceived capability to control the related behavior [8]. To improve the explaining power of TPB, the model has been extended by adding additional important variables, such as locus of control and environmental concern when analyzing pro-environmental behaviors [11,15,16,17]. For instance, environmental concern could exert an indirect influence on consumers’ intention to buy hybrid electric vehicles, and it is positively related to attitude, subjective norm, and perceived behavioral control [11]. Yang and Weber [17] found a positive relationship between internal environmental locus of control and consumers’ pro-environmental behaviors. Among the investigated pro-environmental behaviors, bringing reusable bags for shopping is scarcely analyzed [2]. One exception is the study from Hardy and Bartolotta [15], which conducted an observational experiment to test whether consumers would use reusable bags after getting these bags free. However, studies regarding consumers’ subjective intention towards using reusable bags are seldom conducted. In this regard, this study aims to narrow this research gap by extending the TPB model to uncover the dynamics behind Chinese consumers’ use of reusable bags for shopping. Results of this study could provide valuable insights into Chinese consumers’ use of reusable bags and therefore contribute to plastic crisis management and policy design.

## 2. Literature Review

There are three psychological variables in the framework of TPB including (1) attitude towards behavior, (2) subjective norm, and (3) perceived behavioral control that all combine to influence people’s intention for a related behavior and when the intention becomes strong enough, it turns into actual behaviors [18]. The three basic variables and other two extended variables, i.e., locus of control and environmental concern, are analyzed for their influence on pro-environmental behaviors and intention.

### 2.1. Attitude (ATT)

Attitude refers to an individual’s general positive or negative evaluation of a behavior [18]. In this study, attitude refers to consumers’ evaluation of using reusable bags for shopping. Existing studies have shown that attitude serves as an effective predictor of pro-environmental behaviors [19,20], such as energy saving at workplaces [14] and waste recycling [21]. The more positive the consumers’ attitudes are towards a related behavior, the stronger the intention to conduct the behavior, and vice versa [18].

### 2.2. Perceived Behavioral Control (PBC)

Perceived behavioral control indicates an individual’s perceived difficulty to perform a related behavior [18]. Particularly, the assessment of difficulty depends primarily on external conditions, e.g., preparation time and facilities, and people’s internal controllability and self-efficacy to conduct the behavior [22]. As conducting pro-environmental behaviors may demand some extent of personal inconveniences and sacrifices, some studies identified perceived behavioral control as the most powerful predictor of several pro-environmental behaviors, such as battery pack recycling [23] and waste recycling [21].

### 2.3. Subjective Norm (SN)

Subjective norm refers to an individual’s perception of social pressure from significant referents [18], such as families and friends. Existing studies were divided regarding the role of subjective norm in determining people’s intention for pro-environmental behaviors. Some existing studies identified the significant influence of subjective norm on people’s pro-environmental intention or behaviors, such as household energy saving [24] and PM 2.5 reduction [25]. While other studies found that there was no significant relationship between subjective norm and people’s intention to use renewable energy resources [26]. Generally, it is found that subjective norm exerts more influence in eastern countries emphasizing collectivism than in countries emphasizing individualism [27].

### 2.4. Locus of Control (LC)

Locus of control indicates an individual’s beliefs of whether their individual efforts could produce significant change as a whole [28]. People with internal locus of control believe that their individual behaviors can produce significant changes, and personal behaviors could determine the general outcomes. While people with external locus of control believe individual behaviors exert little influence on the external circumstances [29,30]. It has been shown that consumers who hold external (internal) locus of control are less (more) likely to conduct pro-environmental behaviors [31]. For instance, when consumers ascribe pro-environmental responsibilities to powerful-others (e.g., the government or high technologies), they are less likely to perform pro-environmental behaviors [16], which is also discovered in a study in China [17].

### 2.5. Environmental Concern (EC)

As environmental problems become increasingly alarming in recent decades, people also raise their concern for the severe problems. Technically, environmental concern is defined by Crosby, Gill [32] as the “strong attitude for protecting the environment.” Previous research has highlighted the importance of environmental concern to understand pro-environmental behavior [33]. Consumers with stronger environmental concern are more likely to conduct pro-environmental behaviors [33,34,35]. For instance, environmental concern could indirectly impact people’s pro-environmental behavioral intention by influencing people’s attitude towards the behavior of interest [34], while other studies recognized a direct impact of environmental concern on people’s intention for pro-environmental behaviors [36,37].

## 3. Method

### 3.1. Hypothesis Design

Behavioral intention and actual behavior share the same influential factors, but compared with actual behaviors, behavioral intention is more correlated with the influential factors [30]. Therefore, this study analyzes the influential factors on consumers’ intention to bring a reusable bag for shopping. Based on the existing literature of TPB application in pro-environmental behaviors, five variables are analyzed in this study, including the three basic variables of TPB, i.e., attitude (ATT), perceived behavioral control (PBC), and subjective norm (SN), and two extended variables, i.e., locus of control (LC) and environmental concern (EC).

Attitude in this study measures consumers’ evaluation of using reusable bags for shopping. As this usage is beneficial to plastics reduction, this study infers a positive relationship between consumers’ attitudes and behavioral intention to bring a reusable bag. Perceived behavioral control in this study measures how consumers evaluate their controllability over or necessary resources to bring a reusable bag for shopping. For instance, to bring a reusable bag for shopping, people have to prepare a bag beforehand. Typically, the higher the perceived control over this behavior, the stronger the intention to conduct this behavior. Subjective norm in this study measures consumers’ external pressure from their important people to bring a reusable bag for shopping. For instance, whether consumers’ friends or families have required them to conduct this behavior before. Typically, the higher pressure of the subjective norm, the stronger consumers’ intention to bring a reusable bag for shopping. Based on the above discussions of the three basic TPB variables, this study proposes the following three hypotheses to describe the potential influence of attitude, perceived behavioral control, and subjective norm on consumers’ intention to bring a reusable bag for shopping.

**Hypothesis** **1** **(H1).**
*Attitude positively affects consumers’ intention to bring a reusable bag for shopping.*


**Hypothesis** **2** **(H2).**
*Perceived behavioral control positively affects consumers’ intention to bring a reusable bag for shopping.*


**Hypothesis** **3** **(H3).**
*Subjective norm positively impacts consumers’ intention to bring a reusable bag for shopping.*


Locus of control was added to the basic TPB model in this study to analyze how consumers’ belief of external locus of environmental problem control impacts their intention to bring a reusable bag for shopping. If consumers believe that powerful others are the main promoters of reducing plastics, they are less likely to form an intention to conduct this behavior. Similarly, consumers with external locus of environmental control are also less likely to form positive attitudes towards as well as high perceived behavioral control over this behavior. Therefore, this study proposes the following three hypotheses regarding locus of control.

**Hypothesis** **4** **(H4).**
*External locus of control negatively impacts consumers’ intention to bring a reusable bag for shopping.*


**Hypothesis** **5** **(H5).**
*External locus of control negatively impacts consumers’ attitudes towards bringing a reusable bag for shopping.*


**Hypothesis** **6** **(H6).**
*External locus of control negatively impacts consumers’ perceived behavioral control over bringing a reusable bag for shopping.*


Environmental concern was added in the TPB model to measure how consumers concern about environmental problems caused by the plastic crisis impacts consumers’ intention to bring a reusable bag for shopping. Typically, consumers with higher environmental concern are more likely to conduct this behavior and form a more supportive attitude towards this behavior as well. Accordingly, this study assumes both the direct and indirect influence of environmental concern on behavioral intention and develops the following two hypotheses:

**Hypothesis** **7** **(H7).***Environmental concern positively impacts consumers’ intention to bring a reusable bag for shopping*.

**Hypothesis** **8** **(H8).**
*Environmental concern positively impacts consumers’ attitudes towards bringing a reusable bag for shopping.*


Based on the discussions above, the hypotheses and the research framework of this study are summarized in Figure 1.

### 3.2. Questionnaire Design

The questionnaire consists of three sections. The first section consists of demographic items, e.g., age, gender, education, and income. The second section consists of items regarding the constructs of locus of control and environmental concern. The third section deals with the constructs of TPB, including attitude, subjective norm, perceived behavioral control, and the intention to bring a reusable bag. All items are measured on a five-point Likert scale, with 1 = “completely disagree” and 5 = “completely agree”. Table 1 summarizes the items used for each construct in this study. The questionnaire adds an attention test question (How is the weather today? Please select the yellow color below.) to filter out low quality responses.

### 3.3. Data Collection

This study conducted an online survey from November to December 2021 with a snow-bowling sampling method to investigate consumers’ intention to bring a reusable bag for shopping in China. A pilot study was done with 25 consumers to ensure the items are accurate and understandable to the consumers. Based on the recommendations made by the surveyed consumers, the questionnaire was edited twice before being officially administered. In the questionnaire, the study added an attention test question to test whether the respondents read items attentively or not. The study finally distributed a total of 534 questionnaires, among which 361 were valid after removing those failing the attention test, taking an unusually long time, and with low quality responses. According to 10:1 rule for the ratio of observations to items proposed by Kline [42], the study needs at least 190 valid samples for valid analysis, and the sample size of 361 is enough in this study.

### 3.4. Data Analysis

Descriptive analysis and a Shapiro–Wilk test were conducted with R to summarize the statistical features of designed items and to test the data normality, respectively. Amos 23.0 was used to conduct the Structural Equation Modeling (SEM) and evaluate the correlations and casual relationships among variables.

## 4. Results

### 4.1. Descriptive Summary of Demographics

Table 2 summarizes the descriptive summary of demographic variables. Among the respondents, 60.11% are female and 39.89% are male. The majority of the respondents were born in the 1980s and 1990s, accounting for over 80%. As for education, over 89% of the respondents have an undergraduate diploma or higher. The income of the respondents is distributed evenly, with the percentages for earning less than RMB 5000, RMB 5000–7999, RMB 8000–9999, RMB 10,000–14,999, RMB 15,000–19,999, and over RMB 20,000 are 29.92%, 22.71%, 16.34%, 17.73%, 6.93%, and 6.37%, respectively.

### 4.2. Measurement Model Test: Reliability and Validity

Confirmatory Factor Analysis (CFA) was conducted to assess construct validity of assessments [43]. Based on the initial CFA findings, one item of the perceived behavioral control construct, i.e., PBC4: It is completely up to me whether to bring a reusable bag or not for shopping, has the factor loading as low as 0.40. After removing the item, the CFA was conducted again to evaluate the modified model. Composite reliability (CR) and Cronbach’s alpha were used to evaluate the internal consistency of the items in each construct [44]. As shown in Table 3, both the minimum values of CR and Cronbach’s alpha are higher than the recommended 0.70 [45], suggesting good internal consistency of the items of each construct. Convergent validity was measured by constructs’ factor loadings and the average variance extracted (AVE) [44]. Except for the PBC1 item whose factoring loading value is 0.527, all items have their factor loadings above the recommended threshold value of 0.60 as suggested by Chin, Gopal [46]. The AVE value of each construct is higher than the benchmark value of 0.50 [44]. The results of factor loadings and AVE values indicate good convergent validity of all measurement items.

Discriminant validity was used to evaluate the extent to which two or more constructs should be different from and not related with each other [47]. As shown in Table 4, as the diagonal values, i.e., the square root of each construct’s AVE, are all higher than the correlations of each construct pair, the results are good in discriminant validity [47].

### 4.3. Structural Model Test: Goodness of Fit and Hypotheses Testing

The structural equation modeling was designed to evaluate the correlations among different latent constructs. The SEM model fit is measured by different indices, such as the comparative fit index (CFI), the Tucker–Lewis index (TLI), and the root means squared error of approximation (RMSEA). As suggested by Hair [45] and Kline [42], a model fit is acceptable when the RMSEA value is lower than 0.08 and the CFI and TLI values are higher than 0.90, respectively. Based on the structural equation modeling analysis, $\left(\chi^{2}/df=2.977,GFI=0.897,AGFI=0.861,CFI=0.993,\mathrm{TLI}=0.926,RMSEA=0.074\right)$, the model fit of this study is acceptable. As shown in Table 5, the hypothesis testing results verified the significant influence of consumers’ attitudes $\left(\beta_{H1}=0.364,\;p<0.001\right)$, perceived behavioral control $\left(\beta_{H2}=0.597,\;p<0.001\right),$ and subjective norm $\left(\beta_{H3}=0.228,p<0.001\right)$ on the intention to bring a reusable bag for shopping. Locus of control has no significant impact on consumers’ usage intention for a reusable bag $\left(\beta_{H4}=-0.022,p=0.665\right)$, but this construct could impact intention by exerting a significant influence on attitude $\left(\beta_{H5}=-0.355,p<0.001\right)$ and perceived behavioral control $\left(\beta_{H6}=-0.385,p<0.001\right)$, respectively. While environmental concern does not significantly and directly impact consumers’ intention to bring a reusable bag $\left(\beta_{H7}=-0.005,p=0.916\right)$, the construct could indirectly impact consumers’ intention by impacting consumers’ attitudes towards bringing a reusable bag for shopping $\left(\beta_{H8}=0.423,p<0.001\right)$. Therefore, except for H4 and H7, all hypotheses were supported in this study.

## 5. Discussions

### 5.1. Attitude, Perceived Behavioral Control, and Subjective Norm and the Implications

The study recognized the significant and positive influence of attitude, perceived behavioral control, and subjective norm on consumers’ intention to bring a reusable bag for shopping. In this study, the perceived behavioral control exerts the greatest influence on consumers’ intention to bring a reusable bag for shopping with the highest path weight of 0.597 (see Table 5), which is higher than the path weights of attitude and subjective norm. That is to say, as long as consumers have the available reusable bags and preparation time to bring a reusable bag for shopping, the consumers would intend to conduct this behavior. Some studies also reveal that the perceived behavioral control could exert the greatest influence on other pro-environmental intentions or behaviors within the framework of TPB [21,23,48]. While in other pro-environmental behaviors, perceived behavioral control impacts much less [11,41]. The primary reason for the different impact power of PBC is the varying inconvenience caused by different pro-environmental behaviors. For instance, buying green products does not cause extra efforts during the entire purchasing process. However, to bring a reusable bag for shopping, people have to prepare a bag in advance.

In this study, consumers’ intention to use reusable bags for shopping depends on consumers’ attitudes, which is consistent with other existing findings [41,49]. Moreover, the attitudes towards bringing a reusable bag for shopping is influenced by consumers’ environmental concern and locus of control. In summary, bringing a reusable bag for shopping is a complex decision for consumers. A positive attitude is an important antecedent of pro-environmental behavioral intention.

As for subjective norm, in this study it exerts a positive impact on consumers’ intention to bring a reusable bag for shopping. This finding is consistent with other related studies that also identify the positive influence of subjective norm on consumers’ pro-environmental intention or behaviors [25,27]. While other existing studies find that subjective norm does not necessarily result in consumers’ pro-environmental behaviors [21,49]. The contradicting findings are reasonable as the function of subjective norm varies in different backgrounds [50]. In China, where collectivism culture is emphasized [51], subjective norm may play an important role in determining consumers’ pro-environmental behaviors [27]. However, the influence of subjective norm is only moderate and its average score is only 3.301 (see Table 4), indicating a mild social pressure to bring a reusable bag for shopping. One possible reason is that the majority of Chinese consumers do not have the habit of bringing a reusable bag for shopping by the time of this study, as indicated in an existing study from Wang and Li [6]. Another potential reason is that in some cities where plastic carrier bags are cheap, bringing a reusable bag would be substantially less motivated when the alternative option is costless. Based on the above discussion, policy designers and plastic crisis managers are recommended to help consumers overcome the barriers to bring a reusable bag for shopping and develop the social culture of using reusable bags for shopping. Measures are suggested to the policy designers and plastic crisis managers, such as providing consumers with free initial reusable bags and designing policies to force consumers’ usage of reusable bags, e.g., continuously increasing the prices for plastic carrier bags.

### 5.2. Locus of Control and Intention to Bring a Reusable Bag for Shopping

Results show that locus of control fails to directly impact consumers’ intention to bring a reusable bag for shopping, which is different from those of existing studies that identify the direct influence of locus of control on consumers’ pro-environmental behaviors and intentions [16,17]. However, in this study, this variable indirectly impacts consumers’ intention to bring a reusable bag for shopping by exerting a significant impact on consumers’ attitudes and perceived behavioral control. The average score of external locus of control is only 1.920 (see Table 4), which indicates that the respondents did not believe that individual efforts could not reshape the general results of plastics reduction and environmental protection. However, their intention to bring a reusable bag for shopping is not directly determined by this belief. This finding indicates a gap between consumers’ mentality and their behavioral intention. That is to say, consumers’ mentality has to be converted to a positive attitude or high perceived behavioral control before impacting intention, which has also been verified in other pro-environmental behaviors, e.g., recycling [52]. Once failed, one primary negative consequence is that the mentality would simply stay in people’s minds and lag behind other variables in directly shaping consumers’ intention or behaviors. Unfortunately, most of the consumers may fail to convert this mentality to intention or behaviors. To these consumers, the immediate shopping convenience overshadows their perceived individual influence on plastics reduction and consequently practical obstacles account for more in their behavioral intention.

### 5.3. Environmental Concern and Intention to Bring a Reusable Bag for Shopping

In this study, environmental concern fails to exert a direct influence on consumers’ intention to bring a reusable bag for shopping, and instead it impacts the intention by imposing significant influence on consumers’ attitudes towards this behavior. The results echo the existing findings [11,34], which also identify simply an indirect influence of environmental concern on pro-environmental behavioral intention. Reasons for this finding are twofold. To begin with, there may be a lack of consumers’ perceived benefits of bringing a reusable bag for shopping, such as durable, protecting environment, and beneficial to human health. After all, better understanding or awareness usually precedes positive attitudes towards a product or behavior. For instance, the study by Karaeva, Cioca [53] finds that students with better awareness of renewable energy are more supportive of green energy technologies. Citizens who are insensitive to environmental issues are less likely to engage behavior changes to reduce quantity of municipal solid wastes [54]. In some cases, citizens would develop strong opposition if their perception or attitudes towards waste management facilities are either not well understood or underestimated [55]. During the data collection, a disappointing finding was that not a small number of consumers did not think that environmental problems could be relieved by bringing a reusable bag and believed that changing the materials of plastic bags was the only effective solution. Consequently, it fails to directly motivate consumers’ intention to bring a reusable bag for shopping even they are highly concerned of environmental problems. Just as discussed above, what best motivates consumers’ behavioral intention is the practical convenience, rather than any pro-environmental mentality. The mentality, e.g., environmental concern and locus of control, could only impact indirectly.

## 6. Conclusions

This study conducted a semi-structured online survey on a sample of 361 Chinese consumers to learn consumers’ intention to bring a reusable bag for shopping based on the framework of the theory of planned behavior. The results identify three key findings, which are summarized as follows:

First, all three TPB variables significantly impact consumers’ intention to bring reusable bags for shopping, but the perceived behavioral control impacts the most. The results show that perceived behavioral control ranks first in its path weight to behavioral intention, attitudes second, and subjective norm the least. This finding indicates that the necessary resources (e.g., preparation time) to bring a reusable bag for shopping is the primary determinant of this behavioral intention. Therefore, to encourage consumers’ use of reusable bags for shopping, it is advisable for policy designers and plastic crisis managers to put more effort into making this behavior more controllable or convenient to consumers. Potential efforts include producing easy-to-wash or light-to-carry reusable bags.

Second, locus of control fails to directly impact consumers’ intention to bring a reusable bag for shopping, but it could indirectly impact this intention by influencing consumers’ attitudes towards and perceived behavioral control over this behavior. Namely, attitudes and perceived behavioral control serve as mediators in the relationship between locus of control and consumers’ intention to bring a reusable bag for shopping.

Finally, environmental concern fails to directly impact consumers’ intention to bring a reusable bag for shopping, but it indirectly impacts this intention by influencing consumers’ attitudes towards this behavior. In other words, attitude serves as a mediator in the relationship between environmental concern and consumers’ intention to bring a reusable bag for shopping.

### Research Limits and Future Research

The research is limited by including a small number of old consumers in the survey, which make our findings vulnerable to bias. Future research is directed to recruit a more representative sample to investigate the dynamics behind consumers’ intention to bring reusable bags for shopping in China. In addition, future studies are also encouraged to incorporate actual behavior in the TPB model to analyze the dynamics behind consumers’ behaviors when bringing reusable bags for shopping in China.

## Figures and Tables

**Figure 1 ijerph-19-03638-f001:**
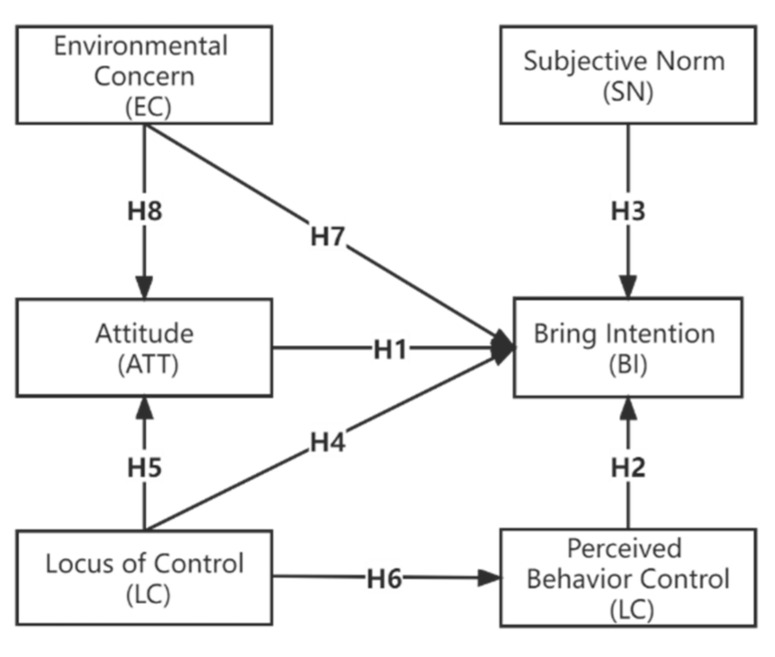
The research framework and hypotheses of this study.

**Table 1 ijerph-19-03638-t001:** Constructs and measuring items of the research model.

Constructs and Measuring Items	Source
Attitude	
ATT1: Bringing a reusable bag for shopping is a good idea.	Sun, Wang [38]
ATT2: Bringing a reusable bag for shopping is necessary.	
ATT3: Bringing a reusable bag for shopping should be encouraged.	
Subjective norm	
SN1: Most people important to me (e.g., families and friends) bring a reusable bag for shopping.	Ru, Qin [25]
SN2: Most people important to me (e.g., families and friends) wish me to bring a reusable bag for shopping.	
SN3: Most people important to me (e.g., families and friends) require me to bring a reusable bag for shopping.	
Perceived behavioral control	
PBC1: If I wish to bring a reusable bag for shopping, then I can do it.	Kim, Njite [39]
PBC2: I have time to prepare a reusable bag before shopping.	
PBC3: I have reusable bags at home that can be used for shopping.	
PBC4: It is up to me whether or not to bring a reusable bag for shopping (deleted for analysis).	
Locus of control	Designed by the definition from Rotter [40]
LC1: In the process of solving environmental problems, individuals cannot exert any influence.	
LC2: In the process of reducing plastics, individual efforts exert limited influence.	
LC3: Reducing plastics should depend on high technologies, rather than individual efforts.	
Environmental concern	
EC1: When humans interfere with nature, it often produces disastrous consequences.	Yadav and Pathak [41]
EC2: The environmental problems become increasingly severe.	
EC3: I feel worried about the environmental problems.	
Bring intention	
BI1: I am willing to bring a reusable bag for shopping.	Sun, Wang [38]
BI2: I plan to bring a reusable bag for shopping.	
BI3: I will try to bring a reusable bag for shopping in the future.	

**Table 2 ijerph-19-03638-t002:** Descriptive summary of demographics.

Items	Range	Frequency	Percentage (%)
Gender	•Female	217	60.11
	•Male	144	39.89
	Total	361	100
Generation	•1950s	3	0.83
	•1960s	11	3.05
	•1970s	33	9.14
	•1980s	100	27.70
	•1990s	192	52.63
	•2000s	24	6.65
	Total	361	100
Education	•Middle school or lower	12	3.32
	•High school	27	7.48
	•Bachelor	176	48.75
	•Master	85	23.55
	•Ph.D.	61	16.90
	Total	361	100
Income	•5000 or less	108	29.92
	•5000–7999	82	22.71
	•8000–9999	59	16.34
	•10,000–14,999	64	17.73
	•15,000–19,999	25	6.93
	•20,000 or more	23	6.37
	Total	361	100

**Table 3 ijerph-19-03638-t003:** Results of measurement model test for reliability and validity.

Construct	Item	Unstd	S.E.	t Value	*p*	Factor Loading	SMC	Cronbach’s Alpha	CR	AVE
ATT	AT1	1.000				0.846	0.716	0.911	0.912	0.775
	AT2	1.090	0.050	21.700	***	0.916	0.839			
	AT3	1.016	0.049	20.867	***	0.878	0.771			
PBC	PBC1	1.000				0.527	0.278	0.745	0.758	0.519
	PBC2	1.241	0.143	8.656	***	0.774	0.599			
	PBC3	1.526	0.182	8.387	***	0.824	0.679			
SN	SN1	1.000				0.751	0.564	0.868	0.872	0.696
	SN2	1.280	0.080	15.977	***	0.941	0.885			
	SN3	1.199	0.078	15.448	***	0.800	0.640			
LC	LC1	1.000				0.797	0.635	0.892	0.894	0.739
	LC2	1.108	0.059	18.856	***	0.938	0.880			
	LC3	0.978	0.055	17.803	***	0.837	0.701			
EC	EC1	1.000				0.692	0.479	0.877	0.879	0.648
	EC2	1.095	0.075	14.594	***	0.870	0.757			
	EC3	1.089	0.075	14.566	***	0.868	0.753			
	EC4	1.069	0.080	13.290	***	0.775	0.601			
BI	BI1	1.000				0.902	0.814	0.913	0.913	0.778
	BI2	1.032	0.046	22.606	***	0.877	0.769			
	BI3	0.980	0.044	22.213	***	0.867	0.752			

Notes: *N* = 361; ATT denotes for attitudes, PBC for perceived behavioral control, SN for subjective norm, EC for environmental concern, LC for locus of control, BI for bringing intention. ***: *p* < 0.001.

**Table 4 ijerph-19-03638-t004:** Descriptive statistics of the constructs and discriminant validity results.

	Mean	SD	AVE	ATT	PBC	SN	LC	EC	BI
ATT	4.379	0.034	0.775	0.880					
PBC	3.941	0.041	0.519	0.203	0.720				
SN	3.301	0.047	0.696	0.098	0.054	0.834			
LC	1.920	0.046	0.739	−0.528	−0.385	−0.141	0.860		
EC	4.227	0.035	0.648	0.568	0.157	0.113	−0.407	0.805	
BI	4.028	0.044	0.778	0.516	0.691	0.298	−0.474	0.329	0.882

Notes: *N* = 361. Diagonal is the square root of AVE from observed variables. Off-diagonal elements are the correlations between different constructs.

**Table 5 ijerph-19-03638-t005:** Summary of hypotheses testing results.

Hypothesis	Path	Standardized Estimate	Estimate	S.E.	C.R.	*p*	Supported
H1	ATT → BI	0.364	0.433	0.074	5.865	***	Yes
H2	PBC → BI	0.597	0.798	0.105	7.630	***	Yes
H3	SN → BI	0.228	0.223	0.044	5.063	***	Yes
H4	LC → BI	−0.022	−0.020	0.046	−0.433	0.665	No
H5	LC → ATT	−0.355	−0.268	0.041	−6.588	***	Yes
H6	LC → PBC	−0.385	−0.258	0.048	−5.366	***	Yes
H7	EC → BI	−0.005	−0.006	0.059	−0.105	0.916	No
H8	EC → ATT	0.423	0.403	0.054	7.439	***	Yes

Notes: *N* = 361. ATT denotes for attitudes, PBC for perceived behavioral control, SN for subjective norm, EC for environmental concern, LC for locus of control, BI for bringing intention. ***: *p* < 0.001.

## Data Availability

The data that support the findings of this study is available upon request to the corresponding author.

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
