# Peer review of "Consumers’ Intention to Bring a Reusable Bag for Shopping in China: Extending the Theory of Planned Behavior"

_ijerph, 2022, doi:10.3390/ijerph19063638_

Round 1

Reviewer 1 Report

Consumers’ intention to bring a reusable bag for shopping in China: Extending the theory of planned behavior

Review comment

This research article evaluates customers behavior towards the use of “reusable shopping bag” through a survey and results were analysed and reported with statistical tool.

Title:

The article title is meaning full and conveys the original meaning and aim of the MS

Introduction:

In the introduction section the need for the study and the theory related to the planned behavior was detailed clearly. But, It is necessary to address following points.

The details related to the previous studies must be addressed in the intro section.

Despite the need for the “reusable bag” author also elucidate the research gap that was identified with the existing literature on this “specific field” in the introduction section

Literature review:

  • Hypothesis design was performed from the literature. However, can it be included in the methodology section?
  • The study framework and hypothesis framed are comparable to other study. (https://doi.org/10.1017/aee.2021.9) Though it is different in location and stat application, authors must take utmost care about this point.

 Methods and results :

  • The methods and results parts are well written and my appreciation to the authors for interpreting the statistical results with a valid reasons.
  • However, in the discussion part, except for the positive criteria, environmental concern and locus control did not posses any impact on the consumer.
  • Page 8 – Line 267 -296: Description related to the Environmental concern and locus control is not sufficient. Authors should add some literature or their observation with subject groups used in this study and provide bit more explanation on the insignificant impact of these parameters.
  • Instead of citing similar research works (Page 8, line – 293), their justification must be referred.

Overall, the manuscripts possess reasonable findings and it can be fine tuned as reported above before the acceptance.

Reviewer 2 Report

REVIEW 1

ijerph-1589698

 Consumers’ intention to bring a reusable bag for shopping in 2 China: Extending the theory of planned behavior

The authors selected a very interesting topic, quite important in the contemporary research on pro-environmental consumer behaviour. The rationale of the study is apparent and literature review supports, to an extent, the hypotheses setting.

However, there are some distinct weaknesses in this manuscript.

Sampling

The snow-bowl technique is a convenience technique that permits any attempt of generalization. The sample is very small and not representative at all. China is an extremely large country and although the specific geographical region of data collection is not mentioned, the authors could and should spend more time in recruiting respondents. After all, this is very easy with online snow-bowling. Moreover, the distributions of gender, age and education are far away from representativeness and thus not acceptable. However, if the sample size was very large, the non-random sampling weakness could be overlooked as weighting of the observed cases could be feasible towards demographical quotas, at least.

Questionnaire

The question for attention test is missing.

Locus of control: Considering that there is rich, extended literature regarding attitudes as an additional background factor to extend TPB, my main incentive to accept this review was the claimed inclusion of locus of control. This might be something original. Unfortunately, the authors chose to approach locus of control by editing some senescence of their own.  In fact, these  look very much like intentions, from a face validity viewpoint. The result was very disappointing.  Discussion provided in 5.2 is very weak, non-convincing if not misleading, I am afraid. 

Data analysis

I am afraid that all statistics are questionable.  

See for example in Table 4 PBC AVE 788/ sq. correlation coeff. 742. This cannot be overlooked as PBC is claimed to be the strongest predictor of intentions. Further, x2/df >3 ; this should be less than 3 in such a small sample. In fact, all GOFs are at the very edge and therefore conclusions cannot be reliable enough.

After all, the effort to expand TPB was not satisfactory, mainly due to unacceptable methodological irregularities. Accordingly, conclusions and implications are not at all satisfactory as there is nothing to add in the so far relevant knowledge, on the broad agenda of pro-environmental consumer behaviour.

The researchers are warmly encouraged to re-design, modify and implement their study taking into consideration the afore mentioned, among others, limitations.

Reviewer 3 Report

The article addresses an interesting topic and tries to explain a specific problem based on a theory with a broad scientific trajectory.

However, the foundation of the research hypotheses in the previous literature that applies the Theory of Planned Behavior (TPB) to the explanation of pro-environmental behavior is somewhat scarce. In particular, the hypotheses that relate the dimensions of locus of control and environmental concern with the TPB variables are poorly supported. The main premise of the TPB is that attitudes, subjective norm and perceived behavioral control determine intention and serve as mediating variables for the influence of other variables. Therefore, it seems logical to suggest that the influence of locus of control and environmental concern on intentions will occur indirectly through the mediation of those TPB variables. Specifically, behavioral control and attitude could serve as mediators for the influence of locus of control and environmental concern, respectively, as they are defined in the paper as related variables of different levels of specify. For instance, the authors state that perceived behavioral control depends on people’s internal controllability (isn’t it internal locus of control?). They also mention that environmental concern is defined in the previous literature as the strong “attitude” for protecting the environment. Thus, one can think that perceived behavioral control-locus of control and attitude-environmental concern are related variables that should interact among them to explain intentions. There is some attempt to do it in H5 and H6, but it is not sufficiently based in a clear definition of variables and relations among them.

Therefore, it is recommended to improve the theoretical foundation of the hypotheses based on other works that relate TPB with pro-environmental behavior and that consider the dimensions of locus of control and environmental concern.

On the other hand, it is recommended to complete the abstract of the article including information on the sample and the data collection and analysis methodology used in the study.

Round 2

Reviewer 1 Report

I thank the authors for implementing necessary corrections effectively.

I can see a lot of improvements in the manuscript

I am satisfied with the modifications

No more corrections required as of my understanding.

Author Response

Thank you for this nice recommendation and this is a big encouragement to us. Thank you for all the time and efforts on our manuscript. We really appreciate it. Best wishes to your life and work! 

Reviewer 2 Report

I am really sorry I have to reject this paper. Revision with regards to the adequacy of sampling as well as variables measurement is not satisfactory. 

Author Response

We are sorry that our revisions fail to be satisfactory. We would make further improvements in our future studies. Thank you for your time and efforts on our manuscript. We really appreciate it.